# Colonization of intervertebral discs by *Cutibacterium acnes* in patients with low back pain: Protocol for an analytical study with microbiological, phenotypic, genotypic, and multiomic techniques

**Vinícius Magno da Rocha**[1]**, Carla Ormundo Gonçalves Ximenes Lima**[2]**, Eliane de Oliveira Ferreira**[2]***, Gabriel Corrêa de Farias**[3]**, Fábio César Sousa Nogueira**[4]**, Luis Caetano Martha Antunes**[5]**, Keila Mara Cassiano**[6]**, Rossano Kepler Alvim Fiorelli**[1]

1 Departamento de Cirurgia geral e especializada, Escola de Medicina, Universidade Federal do Estado do Rio de Janeiro (UniRio), Rio de Janeiro- RJ, Brazil, 2 Departamento de Microbiologia Médica, Institito de Microbiologia Paulo de Góes, Universidade Federal do Rio de Janeiro (UFRJ), Rio de Janeiro-RJ, Brazil, 3 Departamento de Biologia Celular e Molecular, Fundação Oswaldo Cruz, Rio de Janeiro–RJ, Brazil, 4 Instituto de Química, Universidade Federal do Rio de Janeiro (UFRJ), Rio de Janeiro-RJ, Brazil, 5 Centro de Desenvolvimento Tecnológico em Saúde, Fundação Oswaldo Cruz, Rio de Janeiro–RJ, Brazil, 6 Departamento de Estatística, Instituto de Matemática, Universidade Federal Fluminense (UFF), Niterói-RJ, Brazil

* eliane_ferreirarj@micro.ufrj.br

**Data Availability Statement:** No datasets were generated or analysed during the current study. All

## Abstract

Lumbar disc degeneration (LDD) and low back pain (LBP) are two conditions that are closely related. Several studies have shown *Cutibacterium acnes* colonization of degenerated discs, but whether and how these finding correlates with LBP is unknown. A prospective study was planned to identify molecules present in lumbar intervertebral discs (LLIVD) colonized by *C. acnes* in patients with LDD and LBP and correlate them with their clinical, radiological, and demographic profiles. The clinical manifestations, risk factors, and demographic characteristics of participants undergoing surgical microdiscectomy will be tracked. Samples will be isolated and pathogens found in LLIVD will be characterized phenotypically and genotypically. Whole genome sequencing (WGS) of isolated species will be used to phylotype and detect genes associated with virulence, resistance, and oxidative stress. Multiomic analyses of LLIVD colonized and non-colonized will be carried out to explain not only the pathogen's role in LDD, but also its involvement in the pathophysiology of LBP. This study was approved by the Institutional Review Board (CAAE 50077521.0.0000.5258). All patients who agree to participate in the study will sign an informed consent form. Regardless of the study's findings, the results will be published in a peer-reviewed medical journal. Trials registration number NCT05090553; pre-results.

relevant data from this study will be made available upon study completion.

**Funding:** Yes. Conselho Nacional deDesenvolvimento Científico e Tecnológico (CNPq), grant # 310875/2020-0. Fundação de Amparo à Pesquisado Estado do Rio de Janeiro (FAPERJ) grant # E-26/211.554/2019 (Programa Redede Pesquisa em Saúde) and Coordenação de Aperfeiçoamento de Pessoal de NívelSuperior - Brasil (CAPES) - Finance Code 001.

**Competing interests:** The authors have declared that no competing interest exist.

## Introduction

LBP is a common complaint in the general population, and its incidence is about five times higher in patients with LDD and Modic endplate changes [1, 2]. The link between these findings and the isolation of low virulence pathogens in LIVD has already been described in several studies, generating a lot of interest in the topic and promoting the bacterial hypothesis of LDD [3–14].

Albert et al. reported 46% of positive cultures for anaerobes in 61 patients undergoing lumbar discectomy, of which 80% had Modic type 1 changes [15]. Other studies corroborate these findings, with high identification rates for *Cutibacterium acnes*, an anaerobic and aerotolerant pathogen that is bacillus-shaped, stained by the Gram method, pleomorphic, non-spore-forming, and a biofilm producer [11, 16–20]. Rollason et al. isolated strains of this bacterium from the LIVD of 64 patients, revealing the presence of genotypic profiles distinct from those found in the skin, suggesting that specific variants would be related to LBP [21].

Other authors believe that the presence of this pathogen in the LIVD of asymptomatic patients is due to contamination of the specimens during sampling and/or sample treatment in the laboratory environment. Carricajo et al. found positive cultures for *C. acnes* in only 3.4% of the discs from 54 patients, defending the possibility of contamination [22]. Rigal et al. also analyzed LIVD from 313 patients who underwent video-assisted retroperitoneal discectomy, with positive cultures for only six patients [23].

These marked divergences in microbiological results cast doubt on the bacterial hypothesis of LDD and, consequently, raise questions about the possibility of anaerobic pathogens acting as triggers or amplifiers of LBP. Urquhart et al. conducted a systematic review of the literature on the subject and concluded that there is evidence correlating the *C. acnes* isolation with Modic type 1 changes and LBP, but there is a lack of studies producing substantial and irrefutable evidence [24].

Despite the fact that considerable effort has already been expended in determining the presence of *C. acnes* in LIVD, there are still no standard isolation protocols for this pathogen from this clinical specimen. Furthermore, prior to the preparation of this manuscript, studies have shown no consistency in the collection and analysis techniques used, making comparison and reproducibility of results difficult.

Aware of this reality, Astur et al. proposed a protocol for a cohort study to identify *C. acnes* from intervertebral discs [25]. Although valid, the initiative has two significant limitations that should be highlighted. First, after collecting clinical specimens, their immediate inoculation in a liquid medium is not foreseen in the operating room, which increases the contact time of the samples with the aerobic environment and reduces the chances of microorganism recovery even with culture-specific techniques. Second, the authors also highlight sonication (ultrasound waves to release bacteria from the biofilm of surgical prostheses) of specimens for subsequent inoculation in automated blood culture vials; however, this technique was originally described for pathogen recovery from orthopedic prostheses rather than biological tissue [26] and, because the sonication parameters used are not clear (time, power, and frequency), the reproduction of results by other authors and the validation of this protocol is compromised.

Specific microbiological identification techniques, such as immediate inoculation of specimens in liquid culture medium while still in the operating room, vortex for sessile cell recovery, incubation time extension in an anaerobic atmosphere with replication every 72 h, and the use of Matrix-Assisted Laser Desorption Ionization Time of Flight Mass spectrometry—MALDI-TOF MS (Bruker Coorporation), and multiplex-touchdown PCR are strategies that assure reliable results regarding the presence of bacteria in intervertebral discs [27, 28], but they do not explain the pathways through which these microorganisms would participate in LDD and/or LBP.

Once LIVD colonization by *C. acnes* is confirmed using microbiological techniques, in the absence of clinical signs of infection, the main challenge is to understand the implications of this finding for homeostasis of that microenvironment. Rajasekaran et al. showed strong evidence of LIVD colonization by *C. acnes* using rDNA-16s PCR and proteomic analyses [29]. These authors discovered not only host defense proteins, but also proteins associated with bacterial viability and proliferation. Although these findings have demonstrated the active presence of *C. acnes* in intervertebral discs, definitively ruling out the possibility of contamination, they do not necessarily reflect the activity of the expressed proteins, which may remain inactive and without interference in pain-generating pathways and/or the inflammatory response.

In this context, the combined use of proteomic and metabolomic techniques to analyze LIVD colonized by *C. acnes* would be more appropriate in order to assess the subclinical consequences of the colonization by this pathogen [30]. These techniques have already been used successfully to define molecular signatures in other clinical settings [31–34], especially in chronic pain [35–40], and can also provide integrative information on cell function at the molecular level about the role of *C. acnes* in LDD and LBP [41–44].

## Hypotheses and objectives

Our hypothesis is that LDD and LBP are related to LIVD colonization by *C. acnes*.

The primary goal of this study is to determine the incidence of LIVD colonization by low virulent pathogens using specific sampling and culture techniques, as well as phenotypic and genotypic techniques for microbiological characterization.

Other objectives of the study include: 1- assessing the possibility of clinical specimen contamination in positive cultures; 2- establishing a correlation between the presence of the pathogen and the clinical and radiological profile (Modic type 1 changes) of the study participants; 3- to outline a phylotypic profile of the most isolated *C. acnes* strains by bacterial WGS; and 4- identify a molecular profile for LDD and LBP in patients with positive cultures for *C. acnes* through proteomic and metabolomic techniques.

## Justifications

Colonization of LIVD by *C. acnes* in patients with LDD supports the bacterial hypothesis of LBP [4, 8, 29, 45–47]. The determination of a molecular profile for these patients will not only contribute to a better understanding of the pathophysiological basis of LBP associated with LDD [41–43], but it will also positively interfere in the clinical management of this common condition, rationalizing its treatment and optimizing its costs.

Previously studies looked at small groups of patients and did not follow standardized sampling and analysis methods [15, 24, 48, 49]. This makes it difficult to value these findings and while they do suggest a correlation between *C. acnes* and LBP, the evidence is insufficient to explain how this bacterium could be involved in the process of LIVD degeneration and generation/amplification of LBP.

This is the first protocol to investigate the role of *C. acnes* in LIVD degeneration, combining isolation with culture techniques, phenotypic and genotypic characterization techniques; and genomic, proteomic, and metabolomic techniques. In addition, the protocol proposes the clinical–radiological characterization of the participants, blinding the researchers and controlling confounding variables, and following appropriate selection criteria and statistical planning to obtain significant results.

## Methods and analysis

This study protocol is registered with the Research Ethics Committee (REC) [CAAE: 50077521.0.0000.5258] of the Gaffrée and Guinle University Hospital (HUGG) and at *Clinicaltrials.gov* under NCT05090553 (https://clinicaltrials.gov/ct2/results?term=NCT05090553).

### Ethics and dissemination

The protocol will follow the ethical standards of the HUGG REC. All patients whose biological materials will be analyzed will be well-informed of the research objectives and will sign ICF, agreeing with the availability of samples for further laboratory analysis.

### Study design

This analytical study will be performed at a single location (Spine Surgery Center of São Matheus Hospital (HSM) in collaboration with the Gaffrée Guinle University Hospital (HUGG) and Anaerobic Biology Laboratory of Paulo de Góes Microbiology Institute of the Universidade Federal do Rio de Janeiro (UFRJ).

The analyses will be performed sequentially, according to the surgical procedures carried out on the selected participants, taking 6 months after for their recruitment and ending 6 months after the surgical treatment of the last patient. Data from participants will be collected using a form designed specifically for this study.

### Population

Participants included in the study must meet the selection criteria listed below.

**Inclusion criteria.** The inclusion criteria for the study will be participants aged between 18 and 65 years; both sexes; complaint of low back pain lasting more than 3 months; magnetic resonance imaging findings of lumbar disc degeneration (LDD) performed less than 6 months before inclusion in the study; indication for open surgical treatment with isolated microdiscectomy or associated with lumbar arthrodesis; failure of conservative treatment for at least 6 weeks and/or progressive neurological deficit; agreement to follow all phases of the clinical investigation, having signed the informed consent form (ICF) for participation in the study.

**Exclusion criteria.** The exclusion criteria for the study will be a history of open lumbar spine surgery at any stage of life; chemotherapy or pulse therapy with corticoids; immune deficiency; previous intradiscal therapies (nucleotomy or discography); previous endoscopic surgery; history of spinal infection treated with antibiotics in the 6 months prior to inclusion in the study; use of antibiotics in the 2 months prior to the surgical procedure; incomplete research inclusion form; refusal to participate and/or sign the ICF.

### Patient and public involvement

Participant eligibility will be evaluated by the study researchers through a clinical interview and imaging exams evaluation, according to the participants' interest and availability to participate in the study. If the patient agrees to take part in the study, the predetermined selection criteria will be used to evaluate them. The researchers will explain the details of the study and conduct a joint reading of the ICF. Questions about the study objectives, risks and benefits, stages, and research confidentiality will be answered. A patient will only participate upon signing the ICF at this stage of the research. A copy of this document will be given to the participant, another to the responsible researcher, and a third will be attached to the medical record. After signing the consent form, the patient will undergo an evaluation for demographic and clinical data collection and the forms developed for this purpose will be completed. If the

patient is unable to read and sign the written consent form, researchers will verbally explain the study details and the patient will orally provide consent in the presence of a witness who will sign the consent form. The recruitment of participants will take place over a 6-month period, until 120 individuals are included, aiming at a minimum number of 96 collections.

## Participant allocation

The participants will be operated by the same surgical team, and the surgical technique will be posterior discectomy followed by arthrodesis of the approached segment.

**Blinding.** The results of microbiological cultures or molecular analysis will not be disclosed neither to the patients nor to the surgeons. The radiologist who will review the imaging tests will also be blinded to patient data and laboratory results. The researcher who will be analyzing the pain and function scores will be blinded as well.

## Withdrawal of a study participant

A participant will be removed from the study in cases of ICF withdrawal, death, recruitment failure identification, loss of post-surgical follow-up, or the presentation of clinical symptoms of infection, intense pain, fever without other infectious *foci*, increased erythrocyte sedimentation rate and/or C-reactive protein, leukocytosis, imaging tests compatible with spondylodiscitis, or any other condition that leads to blinding interruption.

The reason and circumstances of each participant's withdrawal from the study will be detailed. The data gathered until the patient is withdrawn from the study will be included in the final analysis.

## Selection of outcomes

**Primary outcome.** The main objective of this study will be to determine the incidence of intervertebral disc colonization by *C. acnes* in patients with LBP and LDD. Positive cultures will be used to confirm colonization, followed by phenotypic and genotypic confirmation of the isolated pathogen by mass spectrometry (Biotyper) and rDNA-16s PCR analysis, respectively.

**Secondary outcomes.** *Low back pain*. At the time of patient recruitment the Visual Numeric Scale (VNS) [[50]] will be used to assess LBP intensity and daily activity limitations. A 30% increase in baseline LBP in the first postoperative month will be considered clinically significant. The VNS and the visual analogue scale have a good correlation and are equally sensitive to quantifying postoperative pain [51].

*Quality of life*. The validated Portuguese version of the EuroQol questionnaire (EQ-5D) will be used to assess the quality of life of colonized and non-colonized groups with and without Modic changes [52]. The EQ-5D is a self-administrated standardized instrument containing five items (mobility, self-care, usual activities, pain/discomfort, and anxiety/depression).

*Functionality*. The functionality of the participants will be quantified through the validated Portuguese version of the Oswestry Disability Index (ODI) for LBP [53].

## Host multiomic characteristics

Proteomic and metabolomic techniques will be used to examine disc and plasma samples. The comparison of the group of molecules found in disc and plasma samples may contribute to the quantitative and qualitative identification of profiles using peripheral blood that are associated with the presence of the pathogen in the intervertebral disc. After processing the multiomic raw data using proper bioinformatics tools, statistical techniques will be applied to analyze

them against the clinical–radiological and microbiological profiles of the participants. We expect to provide a molecular signature relating the presence of *C. acnes*, LDD and LBP.

## Adverse effects

Failures related to surgical treatment, such as surgical wound infection, CSF leak, deep vein thrombosis, pain recurrence, or any other adverse events that may arise in the postoperative period, will be reported and considered in data analysis.

## Sampling

The Schedule of enrolment, Interventions, and assessments is shown in Fig 1. During anesthetic induction, intravenous antibiotic prophylaxis will be administered after patient admission to the operating room. Cefazolin at a dose of 1 g replicated every 4 h for the duration of the surgery will be used as an antimicrobial drug. The antibiotic prophylaxis regimen will end with skin closure and will not be extended for additional postoperative time (Fig 1).

The clinical specimens collected will be:

1. Peripheral blood. A 10 mL aliquot will be obtained 30 min after the infusion of surgical antibiotic prophylaxis through upper limb venoclysis. This sample will be placed in a tube containing clot activator for further separation of the serum, which, in turn, will be stored at −80˚C for further analysis.

2. Skin swab. After skin asepsis with 2% chlorhexidine gluconate degerming solution and 0.5% chlorhexidine gluconate alcoholic solution, a swab will be collected from the region where the surgical incision will be performed.

3. Intervertebral disc. Five fragments will be collected at each level approached during surgery. A set of sterile surgical tweezers will be used exclusively for collection, one for each tissue fragment collected. The forceps with tissue fragment will be given to a member of the laboratory who will monitor the procedure inside the operating room immediately after collection. Each fragment will be placed in a vial containing thioglycolate broth (Merck®, Brazil) and 20 glass beads. The vials will be labeled with the initials of the study participants and the type of clinical specimen. After collection, the vials containing the clinical specimens will be kept at room temperature (18˚C to 22˚C) and transported in a specific case to the laboratory within a maximum period of 2 h.

4. Muscle–ligament tissue. Before obtaining the LIVD fragments, five fragments of muscle–ligament tissue adjacent to the collected disc will be collected. The same collection and initial cooling precautions used for the disc fragments will be followed when collecting the adjacent tissue.

### Laboratory analysis

**Cultures.** The skin swabs will be opened only in a laboratory setting. Samples will be seeded on blood agar plates (5% defibrinated sheep blood; blood agar base, 40 g/L; agar, 5 g/L) and anaerobic blood agar (5% defibrinated sheep blood; blood agar base, 40 g/L; agar, 5 g/L; hemin, 10 mL/L; menadione, 5 drops/L. After sowing in solid media, the swab will be inoculated in thioglycolate medium, where it will remain for 14 days. One of the plates will be incubated in a capnophilic atmosphere (5% to 10% $CO_2$), while the other will be incubated in strict anaerobiosis (atmosphere containing 10% $CO_2$, 10% $H_2$, and 80% $N_2$) using an anaerobic chamber or Glove Box (Coy Labs®, USA).

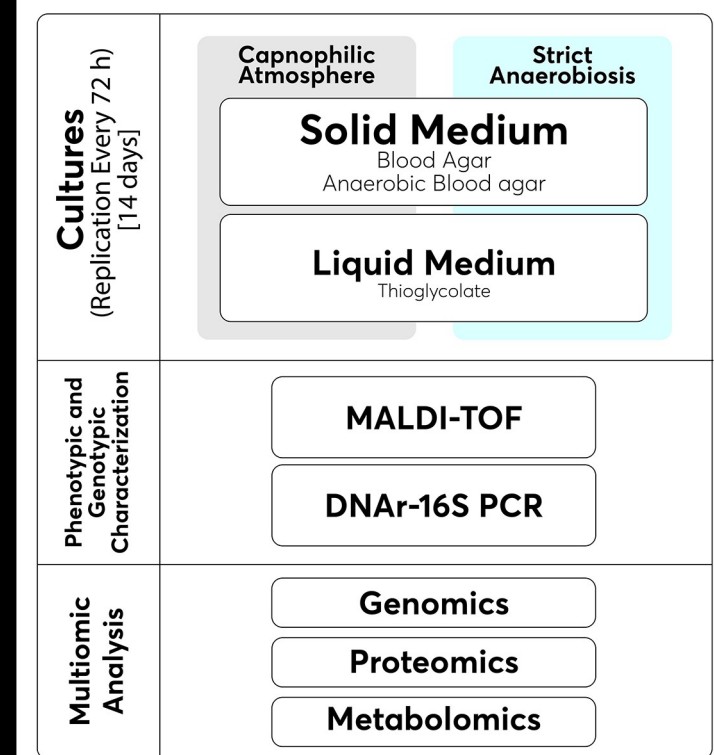

**Fig 1. Flowchart showing all stages of the study.**

The plates will be read after 24 h of incubation and, in case of growth, the colonies will be identified. In the absence of growth, they will be incubated again, and a new reading will be performed after 72 h. The thioglycolate vials containing skin swabs will be kept in a bacteriological incubator (35˚C to 37˚C) for 14 days. In a laboratory setting, the tubes containing disc fragments and muscle–ligament tissue will undergo vortexing (Even EVX2800-BI®, Brazil) for 15 s. All tubes will be kept in a bacteriological incubator (35˚C to 37˚C) and, after 24 h, the first subculture will be carried out using a 100 μL bacteriological loop. An aliquot of the thioglycolate will be taken and plated on blood agar and anaerobic blood agar. The first plate will be incubated in a capnophilic atmosphere and the second in strict anaerobiosis using an anaerobic chamber or Glove Box (Coy Labs®, USA). The plates will be read after 24 h and the colonies will be identified in case of growth. In the absence of growth, they will be incubated again, and a new reading will be performed after 72 h. The same procedure will be used for all thioglycolate vials (totaling six vials for each level approached). Thioglycolate tubes containing the clinical specimens will be kept in an oven for 14 days. Subcultures will be carried out after 72 h and always every 3 days, with the last subculture being carried out on the 14th day of incubation in the same solid culture medium and atmospheres used in the first 24 h.

**Quality control of culture media.**   Sterile saline solution at 0.9% will be used to control the sterility of the culture medium and *Pseudomonas aeruginosa* ATCC 27853 and *Bacteroides fragilis* ATCC 25285 strains to evaluate the recovery capacity of facultative microorganisms and strict anaerobes. Each new batch of culture medium prepared will undergo this quality control.

## Phenotypic characterization

The phenotypic identification of the species will be carried out through mass spectrometry using MALDI-TOF MS equipment (Bruker Biotyper®, Germany*).* The score used for reliable identification of pathogens will follow the manufacturer's instructions.

## Genotypic characterization

The microorganisms isolated from cultures will be subjected to PCR analysis performed in two phases, as described by Barnard et al. [28]. The first to confirm the presence of bacteria (target and non-target) by using species-specific primers targeting the rRNA-16s and a second one, Multiplex-PCR, targeting the virulence and oxidative gens.

## Multiomic analyses

**Whole genome sequencing.**   For the WGS, the DNA of the isolated *C. acnes* strains will be obtained with the Qiagen Blood & Tissue DNA extraction kit (Qiagen®, USA) and subsequent sample purification using RNase.

The extracted and purified DNA will be dosed, and adjusted to a concentration of 0.2 ng/μL. The ends of its two ribbons will be fragmented and tagged (forward and reverse 5'-3'). This will form fragments of different sizes. This data will be used to create a database on the Nextera Flex Kit platform (Ilumina®, USA). The samples will then be multiplexed into a cell stream and run on the NextSeq sequencer (Ilumina®, USA) using paired sequencing to generate files in fastq format files, which will then be filtered using the Trimmomatic tool version 0.36, as recommended by Bolger et al. [54]. This step will be critical for controlling the quality of the analysis because it reduces the possibility of errors during sequencing and allows the use of more reliable reading codes.

The obtained fragments will be sequenced, and their quality will be evaluated with the FastQC tool version 0.11.9. The RedDog software version 1.11 will be used to identify single

nucleotide polymorphisms (SNPs) using the GCA_000008345.1 *C. acnes* reference strain deposited in the GenBank NCBI [https://www.ncbi.nlm.nih.gov/nuccore/] as a comparison. Finally, the PATRIC software version 3.5.21 will be used to create a complete genome phylogeny from the concatenation of SNPs.

**Proteomics.** In this study, proteomics approaches will apply gel-free and in-gel protein digestion by using sodium dodecyl sulfate polyacrylamide gel electrophoresis (SDS-PAGE) technique [2].

Thioglycolate culture mediums will be centrifuged, and the sediment obtained will be washed in phosphate buffer and centrifuged again. The sediment will be dissolved in a lysis solution containing 8 M urea, 2% 3-[(3-cholamidopropyl) dimethylammonio]-1-propanesulfonate, and 40 mM Tris (hydroxymethyl) aminomethane. This lysis will be complemented with the use of glass beads under a Bead Beater agitation (Biospec Product®, USA) (3 × 20 s with intervals of 20 s) followed by cooling for 5 min on an ice bath, as described by Shah et al. [55]. Protein concentration will be measured and 20 μg will be used for both the gel and solution techniques.

The SDS-PAGE -based technique will be used to evaluate the profile of the proteins present in the extract. Proteins will be removed from the gel with a sterile scalpel, eluted, and trypsinized (proteolic digestion) for further analysis through mass spectrometry. In the solution technique, the obtained extract will be trypsinized for further analysis through mass spectrometer. Regardless of the technique used, a nano-Liquid Chromatography coupled to a high-resolution mass spectrometer (nLC-HRMS) will be used.

After analysis by both techniques, the raw data (*.raw*) will be converted into a mascot general file (*.mgf*) using the Mascot Distiller software version 2.8.0, and proteins will be identified using protein banks referring to *C. acnes* and eukaryotes (humans), both created using the NCBI as a *fasta* extension. After this identification, the data will be validated using the Scaffold software version 5 for the same databases. Two biological replicas will be made for both the gel and solution technique. The Interactivenn.net [http://www.interactivenn.net/index.html] [56] and STRING version 11 software [57] will be used for comparative analysis and functional interaction of proteins, respectively.

**Metabolomics.** The metabolomic analysis in this study will use intervertebral disc fragments with and without isolated *C. acnes*. The tubes containing the disc fragments in thioglycolate liquid culture medium will be centrifuged and the sediment will be transferred to 2 mL flat-bottom polypropylene tubes containing a titanium bead (one per tube). The liquid content of these tubes will be evaporated by centrifugation with the CentriVap SpeedVac apparatus (Labconco Corporation®, USA) and then weighed to determine the dry weight. The tubes will then have 500 μL of MS Grade (Fluka) acetonitrile added followed by Bead Beater agitation (Biospec Product®, USA) (4 x 30 s). When all the material has been dissolved, the tubes will be centrifuged, and the supernatant will be stored at −80˚C in new sterile tubes. To identify the metabolites, the material will be injected into a high-resolution mass spectrometer (UHPLC Q-TOF MS/MS). The data obtained will be analyzed in the Metaboanalyst 5.0 database [https://www.metaboanalyst.ca/] and its functions and interactions in the STRING software version 11 [57].

## Questionnaires

Questionnaires on pain (VNS), functionality (ODI), and quality of life (EQ-5D) will be given at recruitment as well as at 1, 3, and 6 months postoperatively. A professional who is not involved in the study will collect all questionaires. Follow-up clinical visits will be carried out at 1, 3, and 6 months postoperatively, with acceptance deviation of 7, 14, and 21 days, respectively.

## Imaging studies

Imaging studies include 1.5 T magnetic resonance imaging. The tests must be dated no later than 6 months prior to the participant's inclusion in the study. Any of the following changes will be considered disc degeneration: disc protrusion, extrusion, or sequestration; reduced disc intensity; annular ruptures; Schmorl's nodes (Depressions on the surface of the vertebral bodies identified on MRI scans that result from the pressure exerted by the intervertebral discs on the endplates, causing intervertebral disc tissue herniation and displacement into the adjacent vertebral bodies); Modic changes; and/or loss of disc height.

## Modic changes

The presence of Modic type 1 changes will be considered in the evaluation of magnetic resonance images. This change is characterized by the presence of a high signal on T2-weighted sequences, with reduced signal on T1-weighted sequences [1, 2]. The frequency of Modic type 1 changes in the participants will be calculated for each of the research groups (presence or absence of colonization) and for the total number of discs analyzed in the laboratory.

## Pfirrmann classification

Degenerative changes identified in the discs will be categorized according to the Pfirrmann classification [58] using T2-weighted magnetic resonance images.

## Confounding variables

As some of this information may change during the study, the data obtained at the time of inclusion in the study will be considered. Below are the confounding variables to be analyzed:

- Age

- Sex

- Education

- Leave of absence

- Alcohol consumption

  1. None or sporadically ($<$ 1 dose/day)

  2. Mild (1–2 doses/day)

  3. Moderate/High ($\geq$ 3 doses/day)

- Smoking

  1. Non-smoker

  2. Smoker

  3. Ex-smoker

- Body mass index

  1. Underweight ($<$ 18.5 Kg/m$^2$)

  2. Normal (18.5–25 Kg/m$^2$)

  3. Overweight (25–30 Kg/m$^2$)

4. Obese ($> 30$ Kg/m$^2$)

- Physical activity practice

  1. Sedentary (not performing physical activity for at least 10 continuous minutes during the week)

  2. Active (vigorous activities for three or more days/week and for 20 min or more per session; or moderate activity or walking for five or more days/week and for 30 min or more per session; or any activity added together 5 or more days/week and 150 min or more/week)

  3. Very active (vigorous activity for five or more days/week and 30 min or more per session; vigorous activity for three or more days/week and 20 min or more per session, more moderate activity and/or walking for five or more days/week and for 30 min or more per session)

- Use of oral steroids up to 3 months before surgery

- Diabetes

## Statistical planning

### Sample size calculation

Considering the main outcome proposed, the identification of disc colonization by *C. acnes*, the minimum sample size (*n*) for the population of interest will be 96 participants, according to the calculation described by Medronho et al., [59] using a confidence interval of 95% and a maximum margin of error of 10%.

The ideal sample size for analysis of secondary outcomes will depend on the incidence of the main outcome. If it is too low, an increase in the number of patients included will be performed with the inclusion of more blocks of participants. In order to have a more accurate dimension, the sample size calculation will be revised as soon as we reach half of the initially planned sample size.

### Statistical analysis methodology

The database will be analyzed using the SPSS software, version 22.0, and R software, version 4.0.2. Descriptive analysis will be performed by charts and descriptive statistics. The inferential analysis will consider a maximum significance level of 5%. The chi-square test or Fisher's exact test will be used for inferential analysis of the distributions of categorical variables. The odds ratio (OR) will be the measure used to estimate risk. The OR's significance will be evaluated by the OR's asymptotic confidence interval. The hypothesis of normality will be verified by the Kolmogorov–Smirnov and Shapiro–Wilk tests. Student's t-test or Mann-Whitney test will be used in the comparison of the two independent groups. More than two independent groups will be compared by the ANOVA or by the Kruskal–Wallis test. Two repeated measures in different assessments will be compared by the paired Student's t-test or by the Wilcoxon's signed-rank test. More than two repeated measures will be compared by the ANOVA for repeated measures, or by the Friedman's test.

The identification of an optimal cutoff point for the marker associated with an outcome will be established through the analysis of the receiver operating characteristic (ROC) curve. The performance will be evaluated by the area under the ROC curve (AUC), and the significance of the AUC will be evaluated by the test that judges the null hypothesis that AUC is equal to 0.5. In addition to the significance test, the asymptotic confidence interval for the AUC will be obtained.

Analysis of the association between two quantitative variables will be visualized by scatter plots and quantified by the Pearson correlation coefficient, in the case of normal distributions, or by the Spearman's correlation coefficient, if normal distribution is not verified in at least one of the variables. The significance of the correlation coefficient will be evaluated by the correlation coefficient test.

Linear or non-linear regression models will be proposed to explain the relationship between the clinical–radiological variables of the patient and the quantitative outcome variables (colonization and multiomic profile). For the variable colonization (binary), logistic regression models will be proposed. Regression model parameters will be estimated by the maximum likelihood method, and the forward Wald method will be used to choose the variables. The goodness of fit of the model will be analyzed by adherence statistics, residual analysis, and verification of theoretical assumptions of the model.

## Supporting information

**S1 Checklist. SPIRIT 2013 checklist: Recommended items to address in a clinical trial protocol and related documents**[*].
(PDF)

**S1 File.**
(PDF)

**S2 File.**
(PDF)

## Acknowledgments

The authors gratefully acknowledge the assistance of all undergraduate and graduate students involved in this study.

## Author Contributions

**Conceptualization:** Vinícius Magno da Rocha, Gabriel Corrêa de Farias, Rossano Kepler Alvim Fiorelli.

**Formal analysis:** Keila Mara Cassiano.

**Funding acquisition:** Eliane de Oliveira Ferreira.

**Investigation:** Vinícius Magno da Rocha, Carla Ormundo Gonçalves Ximenes Lima, Fábio César Sousa Nogueira, Luis Caetano Martha Antunes.

**Methodology:** Carla Ormundo Gonçalves Ximenes Lima, Eliane de Oliveira Ferreira, Gabriel Corrêa de Farias, Fábio César Sousa Nogueira, Luis Caetano Martha Antunes, Keila Mara Cassiano.

**Software:** Fábio César Sousa Nogueira.

**Supervision:** Rossano Kepler Alvim Fiorelli.

**Writing – original draft:** Vinícius Magno da Rocha.

**Writing – review & editing:** Carla Ormundo Gonçalves Ximenes Lima, Eliane de Oliveira Ferreira, Gabriel Corrêa de Farias, Fábio César Sousa Nogueira, Luis Caetano Martha Antunes, Keila Mara Cassiano.

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
