## [Decision Letter · Decision Letter 0]

7 Sep 2022

PONE-D-22-14386Colonization of intervertebral discs by Cutibacterium acnes in patients with low back pain: protocol for an analytical study with microbiological, phenotypic, genotypic, and multiomic techniquesPLOS ONE

Dear Dr. Ferreira,

Thank you for submitting your manuscript to PLOS ONE. After careful consideration, we feel that it has merit but does not fully meet PLOS ONE’s publication criteria as it currently stands. Therefore, we invite you to submit a revised version of the manuscript that addresses the points raised during the review process.

Please see my comments and the reviewers' suggestions below.  Minor revisions to this very good protocol would strengthen the proposed study further.  Looking forward to seeing this work proceed.

We look forward to receiving your revised manuscript.

Kind regards,

D. William Cameron, MD

Academic Editor

PLOS ONE

Journal Requirements:

The authors gratefully acknowledge the Brazilian agencies Conselho Nacional de Desenvolvimento Científico e Tecnologico (CNPq), Fundação Carlos Chagas Filho de Amparo à Pesquisa do Estado do Rio de Janeiro (FAPERJ) and the Coordenação de Aperfeiçoamento de Pessoal de Nível Superior - Brasil (CAPES) - Finance Code 001 for supporting this study. 

The funders had and will not have a role in study design, data collection and analysis, decision to publish, or preparation of the manuscript.

5. Please include a caption for figure 1.

7. We note that the original protocol that you have uploaded as a Supporting Information file contains an institutional logo. As this logo is likely copyrighted, we ask that you please remove it from this file and upload an updated version upon resubmission.

8. We note that the original protocol file you uploaded contains a confidentiality notice indicating that the protocol may not be shared publicly or be published. Please note, however, that the PLOS Editorial Policy requires that the original protocol be published alongside your manuscript in the event of acceptance. Please note that should your paper be accepted, all content including the protocol will be published under the Creative Commons Attribution (CC BY) 4.0 license, which means that it will be freely available online, and any third party is permitted to access, download, copy, distribute, and use these materials in any way, even commercially, with proper attribution.

Therefore, we ask that you please seek permission from the study sponsor or body imposing the restriction on sharing this document to publish this protocol under CC BY 4.0 if your work is accepted. We kindly ask that you upload a formal statement signed by an institutional representative clarifying whether you will be able to comply with this policy. Additionally, please upload a clean copy of the protocol with the confidentiality notice (and any copyrighted institutional logos or signatures) removed.

Additional Editor Comments:

This is a well-written, well informed and well thought out protocol for a timely, and perhaps overdue study of correlation of C. acnes presence, and lumbago due to disc disease, at the point of first surgery. The reviews are well done and should be paid attention to by the authors, as their suggestions will improve the robustness of their findings.

In reading the MS, the authors should remove one double-negative in phrasing, and replace the word incidence with prevalence.

The inclusion of a SPIRIT checklist is appreciated; as this is an observational study, the authors might have used the STROBE reportage guidelines and checklist.

Reviewers' comments:

Reviewer's Responses to Questions

**Comments to the Author**

1. Does the manuscript provide a valid rationale for the proposed study, with clearly identified and justified research questions?

Reviewer #1: Yes

Reviewer #2: Yes

Reviewer #3: Yes

2. Is the protocol technically sound and planned in a manner that will lead to a meaningful outcome and allow testing the stated hypotheses?

Reviewer #1: Yes

Reviewer #2: Yes

Reviewer #3: Yes

3. Is the methodology feasible and described in sufficient detail to allow the work to be replicable?

Reviewer #1: Yes

Reviewer #2: Yes

Reviewer #3: Yes

4. Have the authors described where all data underlying the findings will be made available when the study is complete?

Reviewer #1: Yes

Reviewer #2: Yes

Reviewer #3: Yes

5. Is the manuscript presented in an intelligible fashion and written in standard English?

Reviewer #1: Yes

Reviewer #2: Yes

Reviewer #3: Yes

6. Review Comments to the Author

You may also provide optional suggestions and comments to authors that they might find helpful in planning their study.

Reviewer #1: The central question/controversy in this area concerns the conclusion that any organisms recovered from excised tissues by culture result from skin contamination during sampling. The approach proposed to counter this argument is that proteomic or metabolomic signatures obtained from the tissues might indicate infection, as has been concluded from cited studies. Since no control tissues samples are involved in the study (for example analogous tissues from other surgical sites) reliance is placed on effective skin disinfection prior to surgery. However, viable organisms can remain deep within the skin following pre-operative surgical preparation and these could result in false positive cultures, particularly using the enrichment culture approach in broth before plating. To address this problem, the study protocol proposes swabbing the skin after preparation and before excision. The swab is plated directly onto enriched blood agar without treatment with a neutralizing solution. This omission could give a false idea of the efficacy of skin preparation since residual, active chlorhexidine will be transferred from the skin to the swab and onto the plates. A suitable neutralizer solution such as Dey-Engley broth should be included in the swab transfer tube. Even with addition of this step, there remains concern that skin surface swabs will not pick up viable organisms residing deeper within the skin after preparation and before excision.

PCR methods will be used only to identify and characterize organisms recovered by enrichment culture, no attempt being made to amplify microbial DNA directly from tissues without culture. Whilst this approach will suffer the same problem of possible skin contamination it should at least be considered since it could give some idea of levels of microbial DNA present in the tissues (not possible using the tissue enrichment culture approach).

Reviewer #2: I have read your protocol with great interest, which has the potential to provide an important insights into low virulence infections in lumbar intervertebral discs, and have following optional suggestions based on my experience:

1. Homogenization: Our previous study based on fluorescence in situ hybridization confocal scanning laser microscopy has shown that P. acnes is present deep within intervertebral disc tissue as a biofilm and, as a consequence, it is important that the biofilm is disrupted prior to culture to maximize detection and reduce the possibility of a false-negative result (Capoor 2017). With regard to biofilm disassembly, homogenization has demonstrated the ability to disrupt biofilm deep within the tissue, while sonication has shown demonstrated ability to disrupt biofilm on the surface of implants.

2. Sample size: Two meta-analyses of previous studies addressing infection of intervertebral discs reported a pooled prevalence of bacteria at 34% and 36.2%, respectively with P. acnes as the predominant species (Urquhart 2015, Ganko 2015). An appropriate sample size estimation should therefore be calculated based upon these prevalence rates [Sample size = (1.96^2 x PR (1-PR))/0.05^2); note PR = prevalence rate] (Naing 2006). To ensure that the 95% confidence interval estimate of the proportion positive cases is within 5% of the true proportion, a sample size of approximately 350 cases is necessary.

3. CFU/Gram: Disc fragments for culture should be weighed, placed into a Micro Bag (Seward) containing 4 mL of Viande-Levure medium, and homogenized with a Stomacher 80 (Seward) under aseptic conditions. 100 µL of the resultant homogenate should be used to inoculate Wilkins Chalgren Anaerobic Agar with 7% sheep’s blood and vitamin K (Hi Media Laboratories). An Anaerobic Work Station Concept 400 (Ruskinn Technology) should be used for culture; inoculated plates will be incubated for 14 days at 37 °C under an atmosphere of 80% N2, 10% CO2, and 10% H2. The same amount of the homogenate should be cultured aerobically on Columbia Blood Agar (Oxoid) for 7 days at 37°C to detect aerobic bacteria. Following incubation, bacterial colonies should be counted and the quantity of each colonial morphotype will be expressed as CFU/g of tissue using the Miles and Misra method.

4. qPCR verification of the presence of P. acnes. The human β-globin gene should be included as an internal control to allow assessment of the specimen quality and the nucleic acid extraction as well as the inhibition amplification process. Our experience shows that found that significant number of samples are eliminated due to:

a) inadequate DNA concentration (outside of range 2-60 ng/µL)

b) inadequate human DNA control (Ct of human β-globin gene outside of range 21-27)

c) inconclusive P. acnes template Ct values (32-35)

Reviewer #3: Lines 30, 121, 135: correlate refers to a particular type of analysis. Is this intended (only appropriate for 2 continuous variables), or would “link” or similar be a more appropriate word?

Line 470+ will baseline measures for patient reported outcomes be included in models for future time points?

7. PLOS authors have the option to publish the peer review history of their article (what does this mean?). If published, this will include your full peer review and any attached files.

Reviewer #1: No

Reviewer #2: **Yes: **Manu Capoor

Reviewer #3: No

---

## [Author Response · Author response to Decision Letter 0]

19 Oct 2022

Dear Plos One Academic Editor D. William Cameron,

Concerning the manuscript that was recently submitted to your journal as a protocol and untitled “Colonization of intervertebral discs by Cutibacterium acnes in patients with low back pain: protocol for an analytical study with microbiological, phenotypic, genotypic, and multiomic techniques”, we carefully considered all comments/suggestions/corrections, provided by you and the referee, so the manuscript could fit all reviewers. 

 Herein, we explain how the manuscript was revised based on those comments and recommendations. We extend our appreciation for taking the time and effort necessary to provide such insightful guidance. 

Below are the answers based on the journal requirements:

-Request 

Answer: We apologize for the inconvenience. We did check again all the manuscript’s style so it can meet Plos One style and all the changes were made in the file named marked up copy. 

- Request

Answer: We have changed the Funding information and Financial Disclosure with all the information required. The grant number were also included in the manuscript.

- Request:

The authors gratefully acknowledge the Brazilian agencies Conselho Nacional de Desenvolvimento Científico e Tecnologico (CNPq), Fundação Carlos Chagas Filho de Amparo à Pesquisa do Estado do Rio de Janeiro (FAPERJ) and the Coordenação de Aperfeiçoamento de Pessoal de Nível Superior - Brasil (CAPES) - Finance Code 001 for supporting this study. 

Answer: We have removed the funding support details from the manuscript in the acknowledgment section and included our amended statement in the cover letter as it follows: The funders had and will not have a role in study design, data collection and analysis, decision to publish, or preparation of the manuscript.

- Request

The funders had and will not have a role in study design, data collection and analysis, decision to publish, or preparation of the manuscript. Please include your amended statements within your cover letter; we will change the online submission form on your behalf.

Answer: We have included our amended statement within our cover letter.

- Request

Answer: We have removed from the manuscript, besides from Methods the ethics statements (Line 145)

- Request 

5. Please include a caption for figure 1.

Answer: We have included a caption for the figure (Line 256)

- Request

Answer: We have included caption with the supporting information file.

- Request

7. We note that the original protocol that you have uploaded as a Supporting Information file contains an institutional logo. As this logo is likely copyrighted, we ask that you please remove it from this file and upload an updated version upon resubmission

8. We note that the original protocol file you uploaded contains a confidentiality notice indicating that the protocol may not be shared publicly or be published. Please note, however, that the PLOS Editorial Policy requires that the original protocol be published alongside your manuscript in the event of acceptance. Please note that should your paper be accepted, all content including the protocol will be published under the Creative Commons Attribution (CC BY) 4.0 license, which means that it will be freely available online, and any third party is permitted to access, download, copy, distribute, and use these materials in any way, even commercially, with proper attribution.Therefore, we ask that you please seek permission from the study sponsor or body imposing the restriction on sharing this document to publish this protocol under CC BY 4.0 if your work is accepted. We kindly ask that you upload a formal statement signed by an institutional representative clarifying whether you will be able to comply with this policy. Additionally, please upload a clean copy of the protocol with the confidentiality notice (and any copyrighted institutional logos or signatures) removed.

Answer: The file we have provided and uploaded as a supporting file contains all the details concerning the approval of our study in the Brazilian ethics committee. This pdf file is provided by the website where all the documentation was sent and the study approved, named “Plataforma Brasil”. When the document/receipt is downloaded it comes with the logo from the website. This document that we are sending to the Plos One Journal contains some details of the research, such as the approval number, title and researcher who had the research approved, and where it will be conducted. Unfortunately, the logo can not be removed.. We have provided a caption together with this document in the supplement file.

- Request 

Answer:

We have reviewed the reference list of the manuscript to ensure that it is complete and correct. 

Additional Editor Comments:

This is a well-written, well informed and well thought out protocol for a timely, and perhaps overdue study of correlation of C. acnes presence, and lumbago due to disc disease, at the point of first surgery. The reviews are well done and should be paid attention to by the authors, as their suggestions will improve the robustness of their findings.

- Request

In reading the MS, the authors should remove one double-negative in phrasing, and replace the word incidence with prevalence.

Answer:

The double negative was removed from the phrasing (line 487). The word incidence was replaced in the text by prevalence. 

- Request

The inclusion of a SPIRIT checklist is appreciated; as this is an observational study, the authors might have used the STROBE reportage guidelines and checklist.

Answer: During the article submission a SPIRIT checklist was included. 

Concerning the reviewer’s team collective input, we considered all suggestions/correction as it follows: 

1- Endeavored all comments, suggestions and corrections appointed by the referees;

2- Clarified portions of the text, including introduction, material and methods, results and discussion;

3- Corrected any grammar mistakes found along the text together with miswritten sentences.

# Reviewer 1: 

Reviewer #1: The central question/controversy in this area concerns the conclusion that any organisms recovered from excised tissues by culture result from skin contamination during sampling. The approach proposed to counter this argument is that proteomic or metabolomic signatures obtained from the tissues might indicate infection, as has been concluded from cited studies. Since no control tissues samples are involved in the study (for example analogous tissues from other surgical sites) reliance is placed on effective skin disinfection prior to surgery. However, viable organisms can remain deep within the skin following pre-operative surgical preparation and these could result in false positive cultures, particularly using the enrichment culture approach in broth before plating. To address this problem, the study protocol proposes swabbing the skin after preparation and before excision. The swab is plated directly onto enriched blood agar without treatment with a neutralizing solution. This omission could give a false idea of the efficacy of skin preparation since residual, active chlorhexidine will be transferred from the skin to the swab and onto the plates. A suitable neutralizer solution such as Dey-Engley broth should be included in the swab transfer tube. Even with addition of this step, there remains concern that skin surface swabs will not pick up viable organisms residing deeper within the skin after preparation and before excision.

Answer: Thank you very much for your comments raised concerning the specimen collection. In addition to the tissue collected at each surgical level (five fragments) for the bacterial isolation, another five fragments from the circumjacent tissue are also collected and used as negative control. This sentence will be included in the methodology section (lines 370-371). Concerning the skin disinfection prior surgery with chlorhexidine, we wanted to ensure that the disinfection before pre-operative procedure was effective and that the C. acnes isolated in the intervertebral tissue was not from skin contamination. In fact, previous studies used the same procedure for cleaning the patient’s skin and some were positive and while others were negative. That’s the reason why we decided to use the same protocol.

- PCR methods will be used only to identify and characterize organisms recovered by enrichment culture, no attempt being made to amplify microbial DNA directly from tissues without culture. Whilst this approach will suffer the same problem of possible skin contamination it should at least be considered since it could give some idea of levels of microbial DNA present in the tissues (not possible using the tissue enrichment culture approach).

Answer: We appreciate your comment, but we do not have a species-specific PCR that can be used to identify C. acnes directly from the skin. And even if we did, there are other species with genomes similar to C. acnes, such as the C. namnetense.

Reviewer #2: I have read your protocol with great interest, which has the potential to provide an important insights into low virulence infections in lumbar intervertebral discs, and have following optional suggestions based on my experience:

1. Homogenization: Our previous study based on fluorescence in situ hybridization confocal scanning laser microscopy has shown that P. acnes is present deep within intervertebral disc tissue as a biofilm and, as a consequence, it is important that the biofilm is disrupted prior to culture to maximize detection and reduce the possibility of a false-negative result (Capoor 2017). With regard to biofilm disassembly, homogenization has demonstrated the ability to disrupt biofilm deep within the tissue, while sonication has shown demonstrated ability to disrupt biofilm on the surface of implants.

Answer: Thank you so much for your appreciation. We are totally in agreement. That’s why after the samples are collected, they put in a vial containing thioglycolate broth and with glass beads (approximately 20 beads). When the tubes arrive in the laboratory, they are vortexed vigorously so the tissue can be disrupted (Lines 380-381). 

2. Sample size: Two meta-analyses of previous studies addressing infection of intervertebral discs reported a pooled prevalence of bacteria at 34% and 36.2%, respectively with P. acnes as the predominant species (Urquhart 2015, Ganko 2015). An appropriate sample size estimation should therefore be calculated based upon these prevalence rates [Sample size = (1.96^2 x PR (1-PR))/0.05^2); note PR = prevalence rate] (Naing 2006). To ensure that the 95% confidence interval estimate of the proportion positive cases is within 5% of the true proportion, a sample size of approximately 350 cases is necessary.

Answer: The number of cases was calculated based on the number of surgeries performed at the public hospital where the study will be conducted. This number, according to our calculations, will be 100. As a result, we would most likely have about 400 samples including the swabs from the skin.

3. CFU/Gram: Disc fragments for culture should be weighed, placed into a Micro Bag (Seward) containing 4 mL of Viande-Levure medium, and homogenized with a Stomacher 80 (Seward) under aseptic conditions. 100 µL of the resultant homogenate should be used to inoculate Wilkins Chalgren Anaerobic Agar with 7% sheep’s blood and vitamin K (Hi Media Laboratories). An Anaerobic Work Station Concept 400 (Ruskinn Technology) should be used for culture; inoculated plates will be incubated for 14 days at 37 °C under an atmosphere of 80% N2, 10% CO2, and 10% H2. The same amount of the homogenate should be cultured aerobically on Columbia Blood Agar (Oxoid) for 7 days at 37°C to detect aerobic bacteria. Following incubation, bacterial colonies should be counted and the quantity of each colonial morphotype will be expressed as CFU/g of tissue using the Miles and Misra method.

Answer: We appreciate the description of the protocol, but as a public hospital we do not have a MicroBag or the Viandre-Levure medium. Instead, after collecting all five samples, including controls into thioglycolate with glass beads, tubes will be kept under them room temperature, until they arrive in the microbiology laboratory at the Federal University of Rio de Janeiro (UFRJ) for anaerobic culture. Prior to the first inoculation in blood agar plates, all tubes are vortexed. Plates are incubated under capnophilic and anaerobic conditions for 14 days or until culture growth occurs; tubes are also kept incubated for 14 days or until culture growth occurs. The capnophilic plate is used as a control for contaminants (aerobic bacteria). We will not quantify it, despite the fact that we identify all microorganism grown on plates.

4. qPCR verification of the presence of P. acnes. The human β-globin gene should be included as an internal control to allow assessment of the specimen quality and the nucleic acid extraction as well as the inhibition amplification process. Our experience shows that found that significant number of samples are eliminated due to:

a) inadequate DNA concentration (outside of range 2-60 ng/µL)

b) inadequate human DNA control (Ct of human β-globin gene outside of range 21-27)

c) inconclusive P. acnes template Ct values (32-35)

Answer: Thank you so much for your advice. We considered it during the protocol, but unfortunately, we do not have this goal for the verification of the presence of C. acnes at this time. However, in the future, we plan to conduct several analyses with the C. acnes isolates, one of which will include qPCR for some virulence genes.

Reviewer #3: Lines 30, 121, 135: correlate refers to a particular type of analysis. Is this intended (only appropriate for 2 continuous variables), or would “link” or similar be a more appropriate word?

Answer: Thank you so much for your observation. We think that link or associate would be a better word, because we want to correlate the presence of C. acnes with the patient’s condition and intervertebral discs problems. We have changed the word correlate to associate in lines 29, 122, 133.

Line 470+ will baseline measures for patient reported outcomes be included in models for future time points?

Answer: We appreciate the question. Yes, they probably will.

---

## [Decision Letter · Decision Letter 1]

17 Nov 2022

Colonization of intervertebral discs by Cutibacterium acnes in patients with low back pain: protocol for an analytical study with microbiological, phenotypic, genotypic, and multiomic techniques

PONE-D-22-14386R1

Dear Dr. Ferreira,

We’re pleased to inform you that your manuscript has been judged scientifically suitable for publication and will be formally accepted for publication once it meets all outstanding technical requirements.

Kind regards,

D. William Cameron, MD

Academic Editor

PLOS ONE

Additional Editor Comments (optional):

Reviewers' comments:

Reviewer's Responses to Questions

**Comments to the Author**

1. Does the manuscript provide a valid rationale for the proposed study, with clearly identified and justified research questions?

Reviewer #1: Yes

Reviewer #2: Yes

Reviewer #3: Yes

2. Is the protocol technically sound and planned in a manner that will lead to a meaningful outcome and allow testing the stated hypotheses?

Reviewer #1: Yes

Reviewer #2: Yes

Reviewer #3: Yes

3. Is the methodology feasible and described in sufficient detail to allow the work to be replicable?

Reviewer #1: Yes

Reviewer #2: Yes

Reviewer #3: Yes

4. Have the authors described where all data underlying the findings will be made available when the study is complete?

Reviewer #1: Yes

Reviewer #2: Yes

Reviewer #3: Yes

5. Is the manuscript presented in an intelligible fashion and written in standard English?

Reviewer #1: Yes

Reviewer #2: Yes

Reviewer #3: Yes

6. Review Comments to the Author

You may also provide optional suggestions and comments to authors that they might find helpful in planning their study.

Reviewer #1: The issue of examining "control" tissues has been explained and dealt with satisfactorily.

The thioglycolate broths used for the skin swabs should neutralize any carry-over of chlorhexidine. Primary swabs onto the blood plates may still give false negatives but I note that this has been standard procedure in previous studies.

Reviewer #2: I read reviewer comments and your corrections and believe you did a reasonable and thoughtful job addressing these comments.

Reviewer #3: It would be useful to mention in this text that the baseline measures in the stats analysis section, or at least state a statistician will specify the analyses in a statistical analysis plan.

7. PLOS authors have the option to publish the peer review history of their article (what does this mean?). If published, this will include your full peer review and any attached files.

Reviewer #1: No

Reviewer #2: No

Reviewer #3: No

---

## [Editor Report · Acceptance letter]

13 Dec 2022

PONE-D-22-14386R1 

Colonization of intervertebral discs by *Cutibacterium acnes* in patients with low back pain: protocol for an analytical study with microbiological, phenotypic, genotypic, and multiomic techniques 

Dear Dr. Ferreira:

I'm pleased to inform you that your manuscript has been deemed suitable for publication in PLOS ONE. Congratulations! Your manuscript is now with our production department. 

Kind regards, 

on behalf of

Professor D. William Cameron 

Academic Editor

PLOS ONE